# Balanced Multimodal Learning: An Integrated Framework For Multi-Task Learning In Audio-Visual Fusion

## Abstract

Multimodal learning integrates sensory information from various perspectives, providing significant advantages in different fields like sentiment analysis. However, recent studies have highlighted challenges associated with imbalanced contributions and varying convergence rates across different modalities. Neglecting these imbalances in joint-learning models compromises both information utilization and overall performance. We further find that neither advanced semantic representations nor complex deep networks effectively address these imbalances. To empirically examine these challenges, we approach them from an audio-visual multi-task perspective, focusing on two tasks: lip reading and sentiment analysis, and exploring the contributions of different modalities under varying scenarios. We introduce *BalanceMLA* in our work, a multimodal learning framework designed to dynamically balance and optimize each modality. This framework can independently adjust the objectives of each modality and adaptively control their optimization. Additionally, we propose a bilateral residual feature fusion and an adaptive weighted decision fusion strategy to dynamically manage these imbalances. We also introduce a dynamically generated class-level weighting scheme to cater to fine-grained tasks. Extensive experimental results validate the superiority of our model in addressing modality imbalances, showcasing both its effectiveness and versatility. Furthermore, experiments conducted under extreme noise conditions demonstrate that our model maintains high fusion efficiency and robustness, even in challenging environments.

## 1 Introduction

Human sensory integration, involving vision, auditory, and tactile modalities, offers a rich canvas for perception and understanding. This holistic, multimodal approach has spurred advances in machine learning, particularly in the realm of representation learning. Recent advancements in multimodal learning have not only enhanced the performance of traditional unimodal tasks but also have paved the way for tackling more complex challenges. These include action recognition utilizing temporal and spatial data Xing et al. (2023); Yang et al. (2023), audio-visual fusion in speech processing Zhao et al. (2020); Cheng et al. (2023), and emotion recognition Goncalves & Busso (2022).

However, the pervasive adoption of multimodal learning has also exposed underlying challenges. The predominant concern revolves around modality imbalances—disparities in the contributions of individual modalities to the final decision-making process. Such imbalances not only diminish the utility of less dominant modalities but also create inconsistency in convergence rates, leading to suboptimal model performance Peng et al. (2022); Xu et al. (2023). Existing studies indicate that models employing traditional architectures, such as fully connected layers and Convolutional Neural Networks (CNNs), fall short in dynamically reconciling these imbalances. These models, although robust for unimodal tasks, lack the adaptability required for efficient multimodal integration, thereby reinforcing the issue of modality imbalances.

The advent of transformer-based architectures Vaswani et al. (2017) has revolutionized various domains, including speech, text, and vision, giving rise to powerful models like Wav2vec2.0 Baevski et al. (2020), HuBERT Hsu et al. (2021), BERT Devlin et al. (2018), and Vision Transformer(ViT)

Kolesnikov et al. (2021) for feature learning or encoding. Increasingly, multimodal approaches are leveraging transformer architectures. Despite their inherent advantages—such as self-attention mechanisms, deep network structures, and extensive parameterization—transformer-based models have yet to adequately address the issue of modality imbalances.

In light of these challenges, this work makes several key **contributions**:

- We conduct an in-depth investigation into multimodal models with transformer backbones, focusing on the learning and fusion of features. We analyze the convergence rates, contributions, and fusion efficacy of each modality across various fine-grained tasks. Our findings indicate that the unique self-attention mechanisms, deep structures, and advanced semantic representations in transformers are insufficient to alleviate challenges posed by modality imbalances.

- To address the identified shortcomings, we present a ***balanced** framework for **multi-task learning in **a**udio-visual fusion* (BalanceMLA), a novel framework that dynamically adjusts the learning and optimization process of different modalities. This approach not only enhances overall task performance but also improves the efficiency of feature utilization.

- We introduce two advanced fusion strategies within BalanceMLA: A new bilateral residual feature fusion strategy that seamlessly integrates multimodal features, and an autonomously adaptive decision fusion algorithm, fine-tuning the model's reliance on each modality during inference.

- The framework includes a unique, dynamically generated class-level weighting scheme, making it exceptionally adaptable to tasks that require nuanced semantic understanding.

Our rigorous experimental validation underscores the robustness and versatility of BalanceMLA, establishing its efficacy across various tasks and under challenging environmental conditions.

## 2 RELATED WORK

### 2.1 AUDIO-VISUAL FUSION IN SPEECH TASKS

Emerging studies in audio-visual fusion are making strides in application areas such as speech recognition and speech enhancement. One commonly adopted paradigm is concatenation-based fusion methods Ebrahimi Kahou et al. (2015); Petridis et al. (2018); Afouras et al. (2019), which amalgamate features from both acoustic and visual signals into a unified representation for subsequent neural network-based modeling. However, this approach has the limitation of not explicitly modeling the inter-modal interactions, leaving this crucial aspect to be learned implicitly by the neural network's architecture. Such an implicit representation can be detrimental, particularly when there is a weak correlation between visual cues, such as lip movements, and corresponding acoustic signals.

In speech enhancement tasks, the overarching goal is to elevate the quality of the acoustic signals. Within this framework, visual cues, predominantly lip movements, function as supplementary data for the fine-tuning of acoustic features. Various methods, such as Squeeze-Excitation algorithms Iuzzolino & Koishida (2020), attention-based schemes Sun et al. (2020); Li & Qian (2020), and product-based fusion Wang et al. (2020), are leveraged to achieve this objective. During the training phase, the fused features are meticulously fine-tuned with an emphasis on the acoustic attributes, thus mitigating any potential disruption from the visual modality and optimizing fusion efficacy.

Lip motion patterns are often incorporated with the aim of creating more robust and accurate speech recognition models, especially in noisy conditions. However, this shouldn't imply an exclusive focus on the acoustic modality, relegating visual cues to a secondary, rectifying role. While audio-only models generally exhibit superior discriminative features compared to visual-only models, this performance gap diminishes when visual features are optimized in synergy with acoustic data. Consequently, the performance disparity between the two modalities becomes notably subtle when they participate in specific audio-visual fusion tasks. The optimal balance between these two modalities is precisely what we aims to explore in our work.

## 2.2 AUDIO-VISUAL EMOTION RECOGNITION

In addition to audio-visual speech processing, we also explores audio-visual emotion recognition as one of several fine-grained tasks. Unlike speech tasks, emotion recognition often sees other modalities taking the lead. For instance, when visual, acoustic, and textual modalities are all present, the textual modality often contributes the most to emotion identification. In audio-visual emotion recognition, facial expressions and muscle movements in the visual modality serve as potent cues, often surpassing the expressiveness of pitch and amplitude in the audio modality. However, there are scenarios where facial expressions may be difficult to capture, such as in low lighting or rapid movement. In these cases, discerning specific emotional tones can serve as an effective mechanism for emotion recognition, making audio features critical.

Recent studies have shown combining multiple modalities has proven effective in emotion recognition with appropriate fusion techniques. For example, Goncalves et al. Goncalves & Busso (2022) proposed a deep learning architecture that trains a single-mode auxiliary network and the main network simultaneously to combine and align audio-visual functions.

## 3 METHOD

### 3.1 MODAL IMBALANCE

The issue of modality imbalance has been a persistent challenge in multimodal learning frameworks. Traditional techniques often prioritize one modality over another, based on the assumption that one type of sensory data is more pertinent to a specific task. For instance, the auditory modality may be given more weight in speech recognition tasks, while visual cues may be emphasized in emotion recognition. This modality-centric approach, however, overlooks the potential complementary benefits that could be derived from other less-dominant modalities.

Modality imbalances can also emerge due to the varying quality, availability, and robustness of sensory data. This is particularly true in real-world scenarios, where conditions are not always ideal for capturing every modality uniformly. It can result in a loss of valuable information, affect the model's adaptability, and lead to sub-optimal task performance.

Moreover, the rate at which each modality converges during the training phase can differ Peng et al. (2022), leading to an additional layer of imbalance. This discrepancy may not only result in longer training cycles but also compromise the fused features' quality, thereby affecting the final decision-making process.

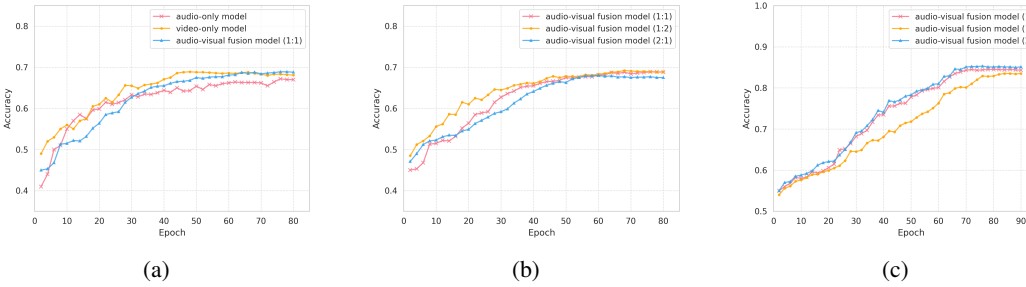

Figure 1: Performance of Transformer Backbone. **(a)** The figure displays the performance of audio-only, visual-only, and audio-visual fusion models in the audio-visual emotion recognition task. Fusion is achieved in a 1:1 ratio, meaning the dimensions of audio and image features are identical, and thus both contribute equally to the task. **(b)** The figure shows the fusion effects in the audio-visual emotion recognition task when the two modalities have varying contributions. The lengths of audio and video features are in the ratios of 1:1, 1:2, and 2:1 respectively. **(c)** The figure illustrates the fusion effects in the audio-visual lip-reading task under varying contribution levels from the two modalities. Here, the lengths of audio and video features are also in the ratios of 1:1, 1:2, and 2:1.

To delve deeper into the issue of modality imbalance in multi-modal tasks, we employ a Transformer architecture with a 6-layer configuration as our backbone for audio-only, visual-only, and audio-

visual fusion scenarios. In the visual modality, we initially utilize a pre-trained RetinaFace model for face recognition and frame-wise cropping. Subsequently, facial image features are encoded using a pre-trained VGGFace2 network, represented as Fv $\in \mathbb{R}^{N_v \times T_v \times L_v}$. For the audio modality, a pre-trained HuBERT model acts as the encoder, yielding features denoted by Fa $\in \mathbb{R}^{N_a \times T_a \times L_a}$, where $N$ is the batch size, $T$ represents the time steps, and $L$ signifies the feature dimensions. To assess the relative contributions of each modality and their subsequent impact on the fusion process, we introduce variability in feature lengths during the fusion stage with a concatenation approach and continuously monitor model performance.

As illustrated in Figure 1a, despite employing high-level semantic features encoded by deep networks, and an acoustic encoding network HuBERT based on Transformer architecture, it's evident that the convergence rates differ under the same learning rate for visual and audio modalities—they converge at epoch 45 and epoch 69, respectively. In emotion recognition tasks, the visual modality tends to outperform the audio modality and contributes more to the overall performance. Conversely, in the lip-reading task, the audio modality significantly surpasses the visual one, suggesting that the dominant modality varies across different multi-modal tasks and contributes more to the specific task at hand.

Figure 1b displays varying lengths of modality inputs to balance the two modalities in emotion recognition tasks. The model's performance shows a minor improvement of approximately 1-2% when the audio input length is twice that of the visual input (represented by the orange line) compared to equal input lengths (the red line). Overscaling the audio input (blue line) may lead to a decline in fusion model performance. Similar trends are observed in the lip-reading task as shown in Figure 1c. However, in this case, increasing the visual input scale improves the model's performance.

These findings confirm the presence of modality imbalance in multi-modal tasks. In fusion networks, dominant modalities can inadvertently impede the learning efficacy of subordinate modalities. We seek to rigorously quantify these disparities by analyzing the input scales across modalities. For example, in emotion recognition scenarios, enhancing the input dimension ratio between audio and visual modalities to 1:2 leads to an improved performance of the fusion model. If the model attains its optimal performance at a ratio $x : y$, we propose that their relative contributions to the task can be approximated as $x : y$.

Addressing these imbalances necessitates the introduction of a dynamic mechanism capable of independently adjusting the contributions and objectives for each modality. This issue is not merely pertinent to specific tasks but has broader implications in the field of multimodal machine learning. This subject is a focal point in our work.

## 3.2 BalanceMLA: Modality Balance in Audio-Visual Fusion

We present BalanceMLA, a balanced multi-task learning framework for audio-visual fusion, designed with a specific focus on audio-visual emotion recognition and lip-reading tasks. As depicted in Figure 2, the architecture comprises two foundational feature extraction modules for both visual and acoustic modalities, in alignment with the techniques delineated in Section 3. These networks independently extract high-level semantic features from raw speech data as well as lip motion or facial expression images. The singular constraint imposed during this phase is that each time-step feature must be encapsulated as a one-dimensional vector.

Our architecture is composed of two semi-independent modules, each tailored for the isolated learning and optimization of distinct modalities. Contrary to being completely disjoint, these modules are interconnected through a sophisticated three-tier fusion mechanism. This mechanism encompasses a novel bilateral residual feature fusion strategy, adaptive weight fusion, and category-level weighting. This multi-faceted fusion scheme fosters effective inter-modality interaction, thereby enriching the quality of the overall multi-modal representation.

**Bilateral Residual Feature Fusion with Cross-Modal Attention (AttentionBRFF).** As previously established, the interplay between visual and acoustic modalities during training can adversely affect the quality of the fused representations, particularly when compared against single-modality benchmarks like speech recognition. Leveraging the simplified residual learning in ResNet He et al. (2016), we introduce residual features as exogenous variables into the respective modality modules.

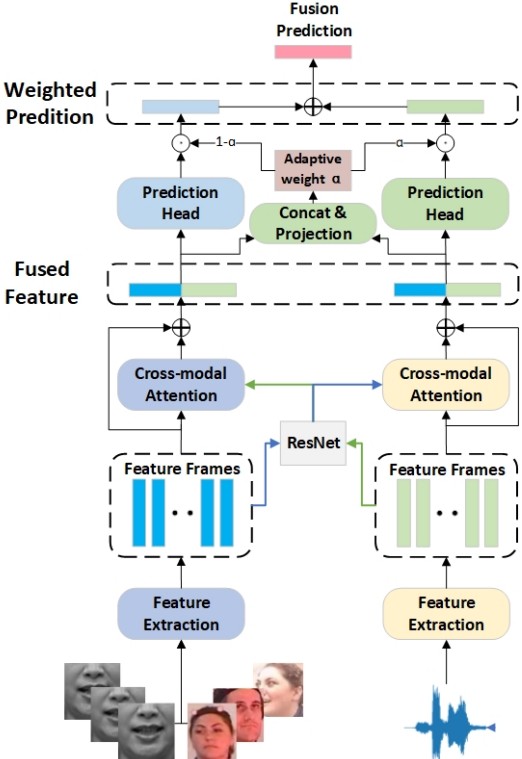

Figure 2: Overview of the BalanceMLA

This allows us to decompose the fusion operation $f_{\text{fuse}} = f_{\text{union}}(f_a, f_v)$ into two sub-operations: $f_{\text{fuse-a}} = \text{AttentionBRFF}(f_a, \text{res}(f_v))$ and $f_{\text{fuse-v}} = \text{AttentionBRFF}(f_v, \text{res}(f_a))$. Here, $f_{\text{fuse}}, f_a, f_v$ denote the high-level features of the fused, acoustic, and visual modalities, respectively. $f_{\text{fuse-a}}$ captures the fusion where the acoustic modality serves as the invariant term, while $f_{\text{fuse-v}}$ operates analogously. This bifurcation approach ensures the retention of essential features from each modality while streamlining the learning of robust fused representations, particularly in cases of low intermodal correlation. It offers a strategic advantage in mitigating issues related to modal imbalance. Within the AttentionBRFF structure, extracted residual features are fed into the complementary modality network. The fusion learning between the two modalities is accomplished using cross-modal attention mechanisms found in Transformer architectures. This internal strategy enhances the capability of our model to adaptively focus on salient features, thereby improving the efficacy of the overall fusion process.

The frame-level residual learning function aims to learn auxiliary information that can fine-tune the features of the basic modality on a frame-by-frame basis. However, acoustic and image features often exhibit mismatched temporal lengths due to differing sampling rates and potential blanks at the data's beginning or end. This creates a challenge in manually establishing correspondence between them. To tackle this issue, we utilize the attention-based method. Specifically, for each time step $j$ in the high-level feature of the basic modality $f_j^{\text{basic}}$, the most relevant residual information is computed using the attention mechanism as:

$$\text{res}(f_j^{\text{aux}}) = \sum_{i=1}^{n} \beta_i f_i^{\text{aux}} \tag{1}$$

where $n$ is the length of the temporal dimension, and $\beta_i$ is the softmax-normalized similarity between $f_j^{\text{basic}}$ and $f_i^{\text{aux}}$, the feature from the other modality. The attention weights $\beta_i$ are calculated as:

$$\beta_i = \frac{\exp\left((f_i^{\text{aux}})^\top W f_j^{\text{basic}}\right)}{\sum_{k=1}^{n} \exp\left((f_k^{\text{aux}})^\top W f_j^{\text{basic}}\right)} \tag{2}$$

This formulation enables the model to automatically pinpoint the most relevant features from the auxiliary modality for each frame in the basic modality, thereby effectively solving the temporal misalignment issue.

**Adaptive decision fusion.** In BalanceMLA, the fused representation obtained after AttentionBRFF processing retains the same shape as its original representation. This compatibility enables the feature to be directly input into the original single-modality model for prediction. At this stage, we introduce a dynamic adaptive decision fusion process, which computes dynamic weights for the features by re-fusing the two output features from the AttentionBRFF mechanism. This adaptivity optimizes the model's performance by selectively emphasizing relevant features, thus enhancing prediction accuracy.

The softmax-normalized weight vector $\alpha$ is generated as:

$$\alpha = \text{softmax}\left(W\left(\left[M(f^{\text{fuse-a}}); M(f^{\text{fuse-v}})\right]\right) + \text{bias}\right)$$

where $M$ is a function that computes the average along the temporal dimension. It enables dynamic weighting of the fused features, allowing for more adaptive and robust decision-making.

**Category level weighting.** Directly weighting the predictions of both acoustic and visual modalities at the model level is suboptimal, as their performances may vary across different categories. To capture the category-specific discriminative capabilities, we concatenate the fused features based on both modalities to dynamically generate weights at the category level. The final fused prediction is then formulated as:

$$\text{result}_{\text{fuse}} = \alpha \odot \text{result}_{\text{audio}} + (1 - \alpha) \odot \text{result}_{\text{video}}$$

where $\text{result}_{\text{fuse}}, \text{result}_{\text{audio}}, \text{result}_{\text{video}}$ are the prediction vectors for the fused, speech recognition, and image classification models, respectively, and $\odot$ represents the Hadamard product. We further compute dynamic weights at the category level, grounded on the existing fusion features, to refine the prediction process. This enables the model to seamlessly switch between various fine-grained tasks with enhanced robustness. Additionally, it effectively mitigates the challenges posed by category imbalance.

## 4 EXPERIMENT

In this section, we comprehensively evaluate our proposed BalanceMLA framework, targeting its performance in sentiment analysis and lip-reading tasks. First, we detail the implementation settings and compare our findings with current state-of-the-art approaches. Following this, we conduct ablation studies to validate the efficacy and underlying design principles of BalanceMLA. During this stage, we perform an in-depth analysis of key components and hyper-parameters to assess their individual contributions to the overall system performance.

### 4.1 EXPERIMENT SETUP

#### 4.1.1 DATASET

In this experiment, we utilize two lip-reading datasets: LRW and LRW1000, and the emotion recognition dataset IEMOCAP.

**LRW:** Yang et al. (2019) The LRW dataset is a large, publicly available English lip-reading corpus featuring short video segments (1.16 seconds) sourced from BBC programs. It encompasses 500 classes with contributions from over 1000 speakers.

**LRW1000:** Yang et al. (2019) The LRW1000 dataset focuses on Mandarin and consists of more than 2000 speakers across 1000 classes. The tasks in LRW1000 present higher complexity than those in LRW due to the larger number of classes and variable video durations, ranging from 0.01 seconds to 2.25 seconds.

**IEMOCAP:** Busso et al. (2008) For emotion recognition, we employ the IEMOCAP dataset, which categorizes emotions into nine types: excitement, happiness, anger, sadness, frustration, neutral, fear, surprise, and other states. In our work, we focus on a four-class benchmark derived from this dataset, featuring a total of 5531 samples. The dataset is partitioned into training, development, and test sets, containing 3871, 553, and 1107 samples, respectively.

### 4.1.2 IMPLEMENTATION DETAILS

For the visual modality, mouth regions are isolated from video frames using the Dlib library King (2009). These extracted mouth images are converted to grayscale and resized to 80x80 pixels, serving as the input for the lip-reading model. For the emotion recognition task, images are processed through the RetinaFace network for facial region extraction. The cropped images are subsequently resized to 128x128 pixels. These processed images are then encoded via a VGG network, yielding high-level image representations with a 1024-dimensional feature space. These features are subsequently transformed to the desired dimensions through a fully connected layer. The same preprocessing is applied to both training and test datasets.

For the acoustic modality, features encoded via HuBERT have a dimensionality of 768. The input speech feature map is constructed as a three-dimensional cube, encapsulating the log-spectrogram, its first, and second-order derivatives. To assess the model's noise resilience, we leverage a speech enhancement technique described in Park et al. (2019), applying varying levels of frequency occlusion, referred to as the mask factor, to the pristine acoustic data.

For the Transformer model employed in Section 3, we set the learning rate to $5 \times 10^{-7}$, utilize 8 attention heads, and construct the architecture with 6 layers. Experiments with 8 and 12 layers yield consistent conclusions, the only difference being initial and final accuracies that deviate by 2% to 5% compared to the 6-layer model. The batch size is set to 16, and we apply a dropout rate of 0.1.

In the case of BalanceMLA, the training learning rate is configured at $3 \times 10^{-4}$ and the fine-tuning learning rate at $5 \times 10^{-5}$. The cross-modal attention mechanism is implemented with 3 layers. The batch size is set to 32, and the entire experimental pipeline is conducted over 30 epochs. The first 25 epochs focus on training, while the remaining 5 epochs are allocated for fine-tuning the model.

We initially train the audio-only (BalanceMLA-AO) and visual-only (BalanceMLA-VO) models independently, followed by fine-tuning the composite models with fusion modules, leveraging pre-trained single-modality models. This process results in models with varied recognition performance, providing insights into their feature representation capabilities.

### 4.2 EXPERIMENTAL RESULTS

We evaluate the performance of our model against various baselines, segmented by the specific tasks under consideration.

**Lip-reading task.** In our experiments, the architecture for the acoustic modality consists of three main components: A spatio-temporal convolution layer with 64 3D convolution kernels of size $5 \times 7 \times 7$, a feature extraction layer utilizing ResNet-34, augmented with a two-layer bi-directional GRU Chung et al. (2014) where each GRU layer is composed of 1024 units, and finally, the 6-layer audio-only transformer model detailed in Section 3. These models are benchmarked against our BalanceMLA-AO framework.

For the visual modality, the experiments comprises the 6-layer visual-only transformer model outlined in Section 3. The model is benchmarked against our BalanceMLA-VO framework.

Based on these single-modal benchmarks, we construct three fusion models: Our proposed method (BalancedMLA), a concatenation-based model inspired by previous work Petridis et al. (2018) (Con), an audio-centric fusion model similar to Sterpu et al. (2018) (Single-Res), and the best-performing fusion transformer discussed in Section 3.

Based on the experimental results in Table 1, the BalanceMLA architecture consistently outperforms other models across both single-modal and multimodal scenarios. For instance, in the audio-only modality on the LRW dataset, BalanceMLA-AO achieves an accuracy of $98.40\%$ and an $F1$ score of $98.33\%$, slightly surpassing the Transformer (audio-only) model, which scores $98.36\%$ and $97.96\%$, respectively. Similarly, on the LRW1000 dataset, BalanceMLA (Fusion) records the highest accuracy and $F1$ score of $86.42\%$ and $85.09\%$, outperforming the nearest competitor, the Transformer (Fusion), with $85.50\%$ and $84.18\%$. These numerical results affirm the efficacy of the BalanceMLA methodology in handling both single-modal and multimodal tasks with superior performance.

**Emotion recognition task.** We selected AVHuBERT Shi et al. (2022), AVBERT Lee et al. (2021), and MAViL Huang et al. (2023) as our baseline models for comparison.

Table 1: Experimental Results on LRW and LRW1000 Datasets

| Model | LRW | | LRW1000 | |
|---|---|---|---|---|
| | Accuracy | F1 Score | Accuracy | F1 Score |
| spatio-temporal convolution layer | 96.70 | 96.53 | 79.86 | 79.82 |
| 2-layer BGRU Chung et al. (2014) | 98.00 | – | – | – |
| Transformer(audio-only) | 98.36 | 97.96 | 83.50 | 83.28 |
| BalanceMLA-AO | **98.40** | **98.33** | **84.32** | **84.05** |
| Transformer(visual-only) | 91.58 | 90.36 | 65.50 | 66.74 |
| BalanceMLA-VO | **93.29** | **93.37** | **66.72** | **65.28** |
| Con (Concatenation) | 97.59 | – | 83.45 | – |
| Single-Res (Audio-Centric) | 97.59 | – | 83.96 | – |
| Transformer(Fusion) | 98.40 | **98.36** | 85.50 | 84.18 |
| BalancedMLA (Fusion) | **98.50** | 98.20 | **86.42** | **85.09** |

Table 2: Comparison with Baseline Models on IEMOCAP Dataset

| Model | Accuracy (%) |
|---|---|
| AVHuBERT (audio-only) | 58.54 |
| AVBERT (audio-only) | 60.94 |
| MAViL (audio-only) | 59.46 |
| BalanceMLA (audio-only) | **66.15** |
| AVHuBERT (visual-only) | 26.59 |
| AVBERT (visual-only) | 45.80 |
| MAViL (visual-only) | 43.03 |
| BalanceMLA (visual-only) | **68.73** |
| AVHuBERT | 46.45 |
| AVBERT | 61.87 |
| MAViL | 54.94 |
| BalanceMLA (Fusion) | **69.32** |

Based on the data presented in Table 2, our BalanceMLA model consistently outperforms baseline models on the IEMOCAP dataset across all experimental settings. For instance, in the audio-only domain, BalanceMLA achieves an accuracy of 66.15%, surpassing AVEBRT, the closest competitor, by a substantial margin of over 5 percentage points. Similarly, BalanceMLA excels in the visual-only scenario with an accuracy of 68.73%, a remarkable improvement over AVEBRT's 45.80%. In the fusion model setup, BalanceMLA further distinguishes itself by attaining an accuracy of 69.32%, setting a new benchmark against all evaluated baselines. These quantitative findings validate the superiority of the BalanceMLA model across distinct modalities.

### 4.3 ABLATION STUDY AND ANTI-INTERFERENCE ANALYSIS

In this section, we investigate the effectiveness of individual components of our proposed model: Bilateral Residual Feature Fusion with Cross-Modal Attention (AttentionBRFF), Adaptive Decision Fusion, and Category Level Weighting. The ablation study is conducted on two challenging datasets, LRW1000 and IEMOCAP.

As shown in Table 3, removing AttentionBRFF leads to a decrease of 2.22% in accuracy, indicating its importance in feature fusion. Omitting Adaptive Decision Fusion also results in a lower accuracy of 83.50%. The importance of Category Level Weighting is also confirmed, as its removal decreases the performance to 82.70%.

The observations from the IEMOCAP dataset, presented in Table 3, are consistent with those on the LRW1000 dataset. The drop in accuracy without AttentionBRFF is 2.22%, demonstrating its signif-

icance. Similarly, without Adaptive Decision Fusion and Category Level Weighting, the accuracy drops to 65.80% and 64.90%, respectively.

Table 3: Ablation Study on LRW1000 and IEMOCAP Datasets

| Comp. | LRW1000 (%) | IEMOCAP (%) |
|---|---|---|
| BaseModel | 82.1 | 64.9 |
| BaseModel + AttentionBRFF | 84.3 | 67.1 |
| BaseModel + AttentionBRFF + Adaptive Decision Fusion | 85.2 | 68.0 |
| FullModel | 86.1 | 69.3 |

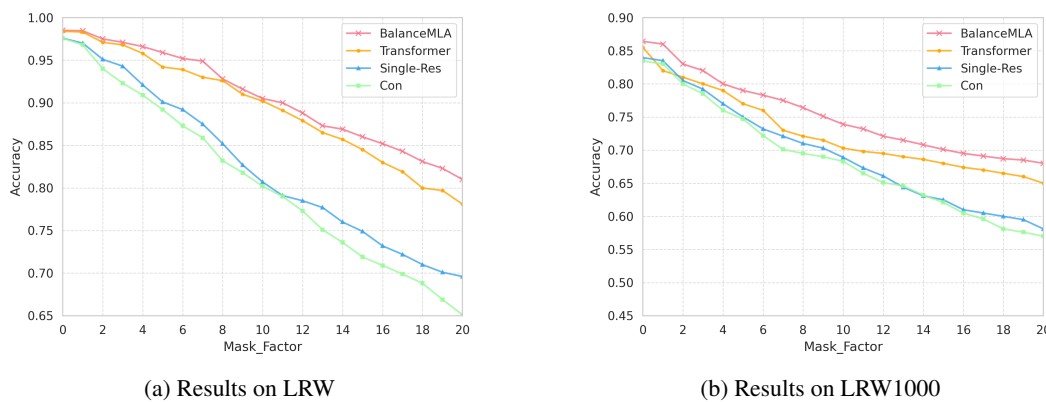

(a) Results on LRW

(b) Results on LRW1000

Figure 3: Anti-interference test of our BalanceMLA

To assess the models' robustness against noise interference, acoustic input was subjected to varying levels of noise contamination. Figure 3 presents the ensuing experimental outcomes. It reveals that, when benchmarked against identical models, the proposed approach demonstrates superior stability across diverse noise environments. This advantage becomes increasingly pronounced as noise levels escalate. Notably, within the LRW1000 dataset, the Single-Res and Concatenation technique shows negligible enhancement in noise resilience, underscoring the imperative of our proposed bilateral fusion strategy. When disparities between the two modalities are considered, Figure 3 confirms that our method consistently outperforms the other fusion models. Furthermore, a positive correlation is observed between the anti-interference capabilities of the fusion model and the recognition performance of the employed benchmark models (like Transformer).

The ablation study and anti-interference analysis confirm the efficacy of each proposed element. Particularly, our model's resilience to acoustic noise highlights its robustness, making it a compelling choice for real-world applications.

## 5 CONCLUSION

In this work, we conducted an in-depth exploration of modality imbalance in the context of transformer architectures. We introduced BalanceMLA, a novel multimodal learning architecture specifically designed for emotion and lip-reading recognition tasks. The architecture integrates three pivotal components: Bilateral Residual Feature Fusion with Cross-Modal Attention (AttentionBRFF), Adaptive Decision Fusion, and Category Level Weighting. Through rigorous evaluation on the LRW, LRW1000 and IEMOCAP datasets, our model consistently outperformed the state-of-the-art baseline models in various experimental settings. The ablation study further reinforced the significance of each integral component. Our future work will be geared towards further optimization of these key elements and the extension of our model to a broader range of multimodal applications.

ETHICS STATEMENT

The focus of this paper is on enhancing the performance of imbalanced representation learning through a novel method. This method and analysis do not raise any ethical issues within the domain of representation learning, as our approach is entirely data-agnostic. However, we caution that while our method improves model performance, it does not inherently mitigate biases present in the training data. As with any representation learning model, BalanceMLA should be rigorously evaluated for fairness and bias before deployment in non-research applications.

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
