# OpenReview forum: "Balanced Multimodal Learning: An Integrated Framework for Multi-Task Learning in Audio-Visual Fusion"
_ICLR.cc/2024/Conference — ICLR 2024 Conference Withdrawn Submission_

### Official Review · Reviewer_jui1 · 2023-10-20

**Soundness:** 2 fair
**Presentation:** 2 fair
**Contribution:** 2 fair
**Rating:** 3
**Confidence:** 4

**Summary:**

This paper is focused on handling modality imbalances in transformer architectures, and propose the BalancedMLA framework, which is a modified version of multimodal transformer architectures. The proposed framework brings out performance improvement on LRW, LRW1000, and IEMOCAP datasets.

**Strengths:**

* This paper analyzes the issue of modality imbalance, especially the audio-visual fusion process in the transformer backbone. These experimental results are inspiring for the community.
* The method of BalancedMLA is clearly introduced, which is also easy to understand.
* The proposed BalancedMLA achieves outstanding performance and impressive improvements on the IEMOCAP dataset.
* The ablation study is meaningful and inspiring for the follow-up works.

**Weaknesses:**

Besides these strengths, the limitations of this paper are also obvious. I hope that my suggestions will be helpful for the authors to polish their paper.
* **Contributions**. This paper mainly focuses on the audio-visual fusion of the transformer backbone, mainly utilized as a fusion network. The significance and contribution of this paper would be a little limited, for which transformers are mainly serving as feature extractors (ResNet in this paper). I think this largely limits their contributions.
* **Technical Design**. The architecture design of BalancedMLA is not novel. It seems that it's adaptive weights based on cross-attention.
* **Experiments**. The improvements on the LRW datasets are very incremental.

Minors:
* There are so many "modality imbalances" in this paper, why the concept is introduced as "MODAL IMBALANCE" in Sec.3.1. I think it's weird.
* It seem that "modality imbalances" in this paper refer to data imbalance, I suggest the authors revise claims for better understanding.

**Questions:**

See Weakness.

---

### Official Review · Reviewer_vtJb · 2023-10-25

**Soundness:** 3 good
**Presentation:** 3 good
**Contribution:** 2 fair
**Rating:** 5
**Confidence:** 4

**Summary:**

The paper proposes a model BalanceMLA to address the issue of modality imbalances in multi-modal tasks such as lip-reading and emotion recognition. The most important parts of the model are AttentionBRFF and an autonomously adaptive decision fusion algorithm. The authors demonstrate that BalanceMLA attains better performance in several datasets compared to some existing models.

**Strengths:**

1. BalanceMLA improves the performance of downstream multimodal tasks.
2. The presentation of the paper is clear and easy to understand.

**Weaknesses:**

1. The experimental comparison seems to be lacking. In the lip-reading task, MixSpeech outperforms AVHuBERT, yet there's no direct comparison presented by the author.
2. The paper's BalanceMLA module bears similarities to the <CATR: Fusion Module for Combination-Dependent Audio Query Converters in Audio-Visual Video Segmentation>. Both utilize a gating mechanism, but the author does not cite CATR. Could you highlight the benefits of BalanceMLA?

**Questions:**

1. Will the author release the code?
2. Can BalanceMLA's performance be verified on the SOTA model, for example, the MixSpeech?

---

> ### Comment · Reviewer_vtJb · 2023-11-23
> **Summary of The Final Review**
>
> Post-rebuttal I think some limitations still exist since necessary experimental comparison seems lacking. So I'm going to keep my original score.

---

### Official Review · Reviewer_iFEs · 2023-10-26

**Soundness:** 2 fair
**Presentation:** 3 good
**Contribution:** 1 poor
**Rating:** 1
**Confidence:** 5

**Summary:**

Modality imbalance usually leads to suboptimal model performance. This paper analyzes the convergence rates, contributions, and fusion efficacy of each modality across various fine-grained tasks of multimodal models with transformer backbones, showing that transformer architecture is insufficient to alleviate challenges posed by modality imbalances. The paper further proposes a balanced framework for multi-task learning in audio-visual fusion, named BalancedMLA. BalancedMLA includes a bilateral residual feature fusion strategy, an adaptive decision fusion algorithm, and a class-level weighting scheme. Experiment results demonstrate the effectiveness of the proposed method.

**Strengths:**

Modality imbalance is a widespread phenomenon in multi-modal learning, balance different modalities in multimodal learning is meaningful.

**Weaknesses:**

- Introduction: The paper says fully connected layers and CNNs reinforce the issue of modality imbalances. But as far as I am concerned, there is no paper proving this. The citation of the related paper should be given to convince me.

- Related work: In the related work part, the paper should introduce recently balanced multimodal learning methods and multimodal multi-task methods.

- Method:
1. The paper says “Figure 1b displays varying lengths of modality inputs to balance the two modalities in emotion recognition tasks. The model’s performance shows a minor improvement of approximately 1-2% when the audio input length is twice that of the visual input (represented by the orange line) compared to equal input lengths (the red line).” However, in Figure 1b, the accuracy of last epoch of audio-visual fusion model (1:1) and audio-visual fusion model (1:2) are almost the same. I can't see an improvement of approximately 1-2%.
2. The proposed AttentionBRFF is not introduced clearly. Why it can streamline the learning of robust fused representations? Why it can enhance the capability of the model to adaptively focus on salient features, thereby improving the efficacy of the overall fusion process?
3. The proposed adaptive decision fusion lacks novelty, it is similar to the fusion method of the paper "On vision features in multimodal machine translation" [1].
4. Why the the proposed method effectively mitigate the challenges posed by category imbalance? It should be explained.
5. In the abstract, the paper says: "we approach them from an audio-visual multi-task perspective," but in the method part, the paper does not introduce how it solves the problem from the multi-task perspective.

- Experiment:
1. On the LRW dataset, the proposed BalancedMLA (Fusion) achieves a lower F1 score than Transformer (Fusion).
2. Lack of experiment results of existing balanced multimodal learning methods, such as OGM-GE and MMCosine, which have been mentioned in the introduction part.
3. Lack of results of multi-task learning.

Reference:

[1] Li, B., Lv, C., Zhou, Z., Zhou, T., Xiao, T., Ma, A., and Zhu, J. On vision features in multimodal machine translation. In Proceedings of the 60th Annual Meeting of the Association for Computational Linguistics (Volume 1: Long Papers), pp. 6327–6337, 2022.

**Questions:**

The paper says "Directly weighting the predictions of both acoustic and visual modalities at the model level is suboptimal." So the paper computes dynamic weights at the category level. How about computing dynamic weights at the sample level?

---

### Official Review · Reviewer_LE8u · 2023-11-03

**Soundness:** 2 fair
**Presentation:** 2 fair
**Contribution:** 2 fair
**Rating:** 3
**Confidence:** 4

**Summary:**

The paper proposes a novel framework called BalanceMLA for balanced audio-visual learning, focusing specifically on emotion recognition and lip reading.
The paper analyzes modality imbalance in audio-visual learning, where one modality dominates and converges faster than others. The paper claims the modality imbalance leads to underutilization of less dominant modalities. To solve modality imbalance, three fusion mechanisms are proposed. Bilateral residual feature fusion outputs multimodal features, adaptive decision fusion combines multimodal features by dynamic weighting, and category-level weighting allows category specific multimodal fusion. Experiments on LRW, LRW1000, and IEMOCAP datasets validate BalanceMLA's effectiveness.

**Strengths:**

- Introduces three key components - bilateral residual feature fusion, adaptive decision fusion, and category-level weighting that together enable balancing modalities. The ablation study validates that each component is necessary for performance improvement.
- Achieves competitive performance on audio-visual datasets like LRW, LRW1000, and IEMOCAP.
- The paper shows the robustness of the model against noise interference.

**Weaknesses:**

- The literature survey is incomplete. The title of the paper is Balanced Multimodal Learning, and the motivation of the paper is to tackle modality imbalance in multimodal learning. However, the literature survey is very limited to only audio-visual speech analysis. Previous studies [1] on multimodal learning for modality imbalance are not discussed. In addition, this work is also relevant to general multimodal learning studies. However, recent studies [2] on multimodal learning are not introduced in the paper. I list a few papers here.

[1] Wang, Weiyao, Du Tran, and Matt Feiszli. "What makes training multi-modal classification networks hard?." Proceedings of the IEEE/CVF conference on computer vision and pattern recognition. 2020.
[2] Nagrani, Arsha, et al. "Attention bottlenecks for multimodal fusion." Advances in Neural Information Processing Systems 34 (2021): 14200-14213.

- The baseline methods are outdated and only a few are compared. There are more recent works on audio-visual learning approaches evaluated on EpicKitchens-100 benchmark [3], Kinectics-400 [4].

[3] Damen, Dima, et al. "Rescaling egocentric vision." arXiv preprint arXiv:2006.13256 (2020).
[4] Kay, Will, et al. "The kinetics human action video dataset." arXiv preprint arXiv:1705.06950 (2017).

- The motivation of the method is unclear. The paper begins by analyzing modality bias by controlling each modality feature dimension. How is feature dimension size related to the contribution of a modality? Is there any reference or rigorous logic or evidence behind this?
Another question is, if the feature dimension is relevant to the contribution of each modality, how is this fact used in the proposed method? The proposed method uses feature fusions, but the size of the feature dimension is not used in the proposed method.

**Questions:**

How is the feature dimension size related to the contribution of a modality? Is there any reference or rigorous logic or evidence behind this?

What is the connection between the modality imbalance introduced in section 3.1 and the proposed method?

The paper proposes a new multimodal architecture that performs well on emotion recognition and lip reading, however, I cannot find a clear connection between the proposed method and modality imbalance.